# The Mechanisms of PD-L1 Regulation in Non-Small-Cell Lung Cancer (NSCLC): Which Are the Involved Players?

**DOI:** 10.3390/cancers12113129

**Published:** 2020-10-26

**Authors:** Giuseppe Lamberti, Monia Sisi, Elisa Andrini, Arianna Palladini, Francesca Giunchi, Pier-Luigi Lollini, Andrea Ardizzoni, Francesco Gelsomino

**Affiliations:** 1Department of Experimental, Diagnostic and Specialty Medicine, S. Orsola-Malpighi University Hospital, Alma Mater Studiorum University of Bologna, Via Massarenti 9, 40138 Bologna, Italy; giuseppe.lamberti8@unibo.it (G.L.); monia.sisi@studio.unibo.it (M.S.); elisa.andrini2@studio.unibo.it (E.A.); andrea.ardizzoni@aosp.bo.it (A.A.); 2Laboratory of Immunology and Biology of Metastasis, Department of Experimental, Diagnostic and Specialty Medicine (DIMES), University of Bologna, viale Filopanti 22, 40126 Bologna, Italy; arianna.palladini@unibo.it (A.P.); pierluigi.lollini@unibo.it (P.-L.L.); 3Laboratory of Oncologic Molecular Pathology, S.Orsola-Malpighi Teaching Hospital, University of Bologna, Via Massarenti 9, 40138 Bologna, Italy; francesca.giunchi@aosp.bo.it; 4Oncologia Medica, Azienda Ospedaliero-Universitaria di Bologna, Via Albertoni—15, 40138 Bologna, Italy

**Keywords:** PD-L1, PD-1, non-small-cell lung cancer, immunotherapy, immune checkpoint inhibitors, T-cell

## Abstract

**Simple Summary:**

Immunotherapy against PD-1/PD-L1 dramatically improved outcomes in non-small cell lung cancer patients. These treatments are more effective the higher the expression of PD-L1 on tumor cells, reported as tumor proportion score. However, PD-L1 expression can be highly variable, depending on different mechanisms of regulation. These mechanisms are usually grouped in intrisc (including genetic and epigenetic factors) and extrinsic factors (i.e., deriving from interaction of tumor cells with tumor microenvironment or other external factors). We reviewed mechanisms underlying PD-L1 expression regulation in order to provide a comprehensive overview and identify key regulatory factors that are or can potentially be exploited to improve outcomes on immune checkpoint inhibitors targeting the PD-1/PD-L1 axis.

**Abstract:**

Treatment with inhibition of programmed cell death 1 (PD-1) or its ligand (PD-L1) improves survival in advanced non-small-cell lung cancer (NSCLC). Nevertheless, only a subset of patients benefit from treatment and biomarkers of response to immunotherapy are lacking. Expression of PD-L1 on tumor cells is the primary clinically-available predictive factor of response to immune checkpoint inhibitors, and its relevance in cancer immunotherapy has fostered several studies to better characterize the mechanisms that regulate PD-L1 expression. However, the factors associated with PD-L1 expression are still not well understood. Genomic alterations that activate *KRAS*, *EGFR*, and *ALK*, as well as the loss of *PTEN*, have been associated with increased PD-L1 expression. In addition, PD-L1 expression is reported to be increased by amplification of *CD274*, and decreased by *STK11* deficiency. Furthermore, PD-L1 expression can be modulated by either tumor extrinsic or intrinsic factors. Among extrinsic factors, the most prominent one is interferon-γ release by immune cells, while there are several tumor intrinsic factors such as activation of the mechanistic target of rapamycin (mTOR), mitogen-activated protein kinase (MAPK) and Myc pathways that can increase PD-L1 expression. A deeper understanding of PD-L1 expression regulation is crucial for improving strategies that exploit inhibition of this immune checkpoint in the clinic, especially in NSCLC where it is central in the therapeutic algorithm. We reviewed current preclinical and clinical data about PD-L1 expression regulation in NSCLC.

## 1. Introduction

Programmed death-1 (PD-1) is a type I transmembrane protein expressed on the surface of antigen-stimulated T cells on which it exerts an inhibitory effect when binding with its ligand programmed-death ligand 1 (PD-L1), which can be expressed by normal cells, immune cells and tumor cells [1]. PD-1 and PD-L1 are referred to as immune checkpoints since they modulate the adaptive immune response to avoid an exaggerated activation and thus play a key role in immune homeostasis. Given its crucial role, PD-L1 expression on the cell membrane is finely regulated at different levels. These involve multiple factors that are classified as intrinsic (i.e., originating within the cell) and extrinsic (i.e., determined by factors outside of the cell), which modulate constitutive and inducible PD-L1 expression, respectively. 

However, PD-L1 can be aberrantly expressed on tumor cells, allowing them to escape immune surveillance and consequent killing by the host adaptive immune system. Targeting PD-1 or PD-L1 with specific immune checkpoint inhibitors (ICIs) to prevent their interaction yielded dramatic results and improved outcomes in several hard-to-treat cancers, including melanoma, urothelial carcinoma, small-cell and non-small-cell lung cancer (NSCLC) [2,3,4,5,6,7,8,9,10,11,12,13]. In NSCLC, PD-L1 expression reported as the percentage of viable tumor cells showing membranous staining for PD-L1 at immunohistochemistry (tumor proportion score, TPS) is a predictive biomarker of response to ICIs targeting PD-1 or PD-L1. Furthermore, higher PD-L1 expression levels in NSCLC are associated with improved outcomes on ICI targeting PD-1 [4,14,15]. Nevertheless, PD-L1 expression on tumor cells is highly variable and different expression levels are associated with different clinicopathological and genomic characteristics in NSCLC [16]. Understanding the mechanisms that underlie PD-L1 expression in the oncology perspective will help in developing novel strategies to improve outcomes on ICIs, by either delaying or overcoming the onset of resistance to this class of drugs through increasing or re-enabling PD-L1 expression on tumor cells or preventing PD-L1 downregulation. 

In this review, we will summarize the molecular factors that affect PD-L1 expression in NSCLC and discuss their potential translational clinical applications.

## 2. Intrinsic Factors 

Genetic and epigenetic alterations that affect constitutive PD-L1 expression on tumor cells are referred to as intrinsic factors (Figure 1). 

### 2.1. Genetic Factors

#### 2.1.1. RAS and Mitogen-Activated Protein Kinase

Kirsten Ras Sarcoma Viral Oncogene Homolog (*KRAS*) harbors activating mutations in about a third of lung adenocarcinomas [17]. Activated RAS induces mitogen-activated protein kinase (MAPK) activation, which in turn inhibits tristetraprolin (an adenylate-uridylate-rich element-binding protein) by p38-dependent phosphorylation. Furthermore, activated RAS stabilizes PD-L1 mRNA. Both mRNA stabilization and tristetraprolin inhibition result in increased PD-L1 expression in lung cancer cell lines [18,19]. This is consistent with the decrease in PD-L1 expression in both mouse and human KRAS-mutant lung cancer cells observed with concomitant Mitogen-activated ERK kinase (MEK) and extracellular signal-regulated kinase (ERK) inhibition [18,19,20]. On the other side, MEK inhibition alone (i.e., without concomitant ERK inhibition) leads to increased PD-L1 expression, possibly because of feedback mechanisms that induce paradoxical activation of the MAPK pathway [21]. 

#### 2.1.2. Receptor Tyrosine Kinases

Upstream of KRAS, the epidermal growth factor receptor (*EGFR*) activation by its ligand binding (EGF) induces PD-L1 expression on NSCLC cells through the phosphatidylinositol 3-kinase-Protein kinase B-mammalian target of rapamycin (PI3K/Akt/mTOR) and the Janus kinase-signal transducer and activator of transcription (JAK/STAT) pathways [22]. *EGFR* is mutated in about 10–25% of lung adenocarcinomas and is associated with female sex, absence of smoking history and Asian ethnicity [23,24,25]. In *EGFR*-mutant NSCLC, EGFR can increase PD-L1 expression through activation of several pathways, including MAPK [26,27], PI3K/Akt/mTOR [27,28], and JAK2/STAT1 and STAT3 in the JAK/STAT pathway [29,30]. Despite the association with a high PD-L1 expression in a small cohort of resected NSCLC specimens [31], the presence of activating *EGFR* mutations is associated with low expression of PD-L1 on tumor cells, as well as a poor outcome to ICIs [16,32,33,34]. Nonetheless, this effect might be depending on the type of *EGFR* activating mutation [35,36]. This apparent paradox can be explained by the complex interaction between tumor cells and the tumor microenvironment (TME). In fact, *EGFR*-mutant cancer cells tend to have a lower tumor mutational burden (TMB), defined as the total number of nonsynonymous mutations per coding area of a tumor genome (Mut/Mb) [37], and to be less immunogenic compared to *EGFR* wild-type tumors. In this context, the deficient interaction between *EGFR*-mutant tumor cells and host immune system cells leads to low T-cell infiltrate which may impair PD-L1 inducible expression [38,39,40].

Similarly, the presence of an anaplastic lymphoma kinase (*ALK*) rearrangement in NSCLC is associated with high PD-L1 expression via activation of the MEK/ERK, PI3K/Akt/mTOR pathways and STAT3 [41,42,43]. The latter can increase PD-L1 transcription by directly binding the promoter region of the *CD274* gene (located on chromosome 9 in the 9p24.1 locus and encoding PD-L1) [43].

#### 2.1.3. PI3K/Akt/mTOR

The PI3K/Akt/mTOR pathway is central in cell homeostasis maintenance and metabolism regulation and is altered in several cancer types, including NSCLC [44,45]. In this tumor type, PD-L1 expression is increased by PI3K/Akt/mTOR activation mediated by either Phosphatase and tensin homolog (PTEN) inactivation or loss, or mTOR activation [28,46,47]. Activated mTOR acts at the post-transcriptional level by recruiting PD-L1 transcripts to active polysomes so that the PD-L1 protein level is increased without a corresponding significant elevation in mRNA levels [20]. Accordingly, tumor suppressor candidate 2 (TUSC2) overexpression determines reduced PD-L1 expression by impairing effect of EGFR, mTOR and Akt activation on PD-L1 expression [48,49]. 

#### 2.1.4. Tumor Suppressors

Recently, the role of tumor suppressors in tuning PD-L1 expression in NSCLC has been widely explored and has gained attention thanks to implications with outcomes to ICIs. Preserved p53 function reduces PD-L1 expression via miR-34 [50]. On the contrary, deleterious mutations of *TP53*, which encodes p53, or aberrant p53 expression, which reflects protein loss of function, are associated with increased PD-L1 expression [51,52] and immune infiltrate, resulting in improved outcome to ICIs [51]. These effects were amplified in NSCLC with co-occurring *TP53* and *KRAS* mutations. A similar association between high PD-L1 expression and p53 overexpression was observed in resected primary lymphoepithelioma-like carcinoma of the lung, a rare type of NSCLC [53]. 

Serine/threonine kinase 11 (*STK11*, or liver kinase B1, *LKB1*) is a tumor suppressor that is often found mutated in lung adenocarcinoma, especially in presence of co-occurring *KRAS* mutations. Mutation or loss of *STK11* is associated with low or absent PD-L1 expression, in both *KRAS*-mutant and *KRAS*-wild type lung adenocarcinoma, and with impaired efficacy of ICIs targeting the PD-1/PD-L1 axis [16,54]. This can be explained by the fact that *STK11* deletion in *KRAS*-driven lung adenocarcinoma promotes the accumulation of neutrophils with T-cell suppressive function, as observed in mouse models [55]. Moreover, an immune “cold” TME can be responsible for the absent or low PD-L1 expression associated with *STK11* mutation or loss in NSCLC [55]. Concurrent *PTEN* and *STK11* loss favors, in vivo, the growth of squamous NSCLC with high PD-L1 expression [56], while lung tumors with adenocarcinoma histology and high PD-L1 expression can be observed in mice where *PTEN* is simultaneously lost with Kelch-like ECH-associated protein 1 (KEAP1), rather than *STK11* [57]. 

DNA damage response and repair (DDR) genes are a large family of genes that encode for caretaker tumor suppressors proteins, such as breast cancer type 1 and 2 susceptibility proteins (*BRCA1* and *BRCA2*), ataxia telangiectasia Mutated (*ATM*), *ATM*-related protein (*ATR*) and many others [58]. Alterations in DDR genes determine the accumulation of cytosolic DNA that induces PD-L1 expression in response to activation of the stimulator of interferon genes (*STING*) [58]. DDR mutations have been associated with increased TMB and improved outcomes to ICIs in NSCLC patients so that combination strategies of DDR inhibitors and ICI are under evaluation in these patients [59,60]. PD-L1 expression is increased in response to *STING* activation thanks to type I interferon response 3 (IRF3) directly binding to *CD274* promoter [58]. Interferon-gamma produced by activated T cells binds to its receptor on tumor cells and determines STAT1 activation and transcription of IRF1/7 which in turn binds to the *CD274* promoter, increases PD-L1 transcription and, ultimately, the expression on tumor cells [61]. 

The *CD274* promoter is the target of several transcription factors since a fine-tuning of PD-L1 expression on tumor cells is needed. Casey et al. showed that Myc can directly bind the *CD274* promoter in several cancer cell lines, including NSCLC cell line H1299 [62]. Consistently, an inverse correlation has been observed between PD-L1 and bridging integrator-1 (BIN1) expression, since the latter inhibits Myc- and EGFR-MAPK-mediated induction of PD-L1 expression [63]. Furthermore, several other transcription factors regulate PD-L1 expression by directly binding the *CD274* promoter or other regulating regions. PD-L1 expression is regulated by NF-κB through the RELA/p65-MUC1-C complex [64], as well as by c-Jun and the activator protein-1 (AP-1), which binds the first intron of *CD274*, in a STAT3-dependent way in *KRAS*-mutant NSCLC [20]. Yes-associated protein 1 (YAP) and WW domain-containing transcription regulator 1 (TAZ) are key components of the Hippo pathway which is commonly deregulated in lung cancer and whose activation determines the upregulation of PD-L1 expression [65,66]. The positive correlation of expression of YAP and PD-L1 determined by immunohistochemistry in human NSCLC cells corroborates this [67]. The transcriptional enhancer factor 1 (TEF-1) is a downstream effector of the Hippo pathway and also directly binds the *CD274* promoter enhancing PD-L1 expression in lung cancer cell lines [66,67,68,69]. As a consequence, upstream kinases and inhibitors of the Hippo pathway, mammalian STE20-like kinase 1 and 2 (MLST1/2) and large tumor suppressor 1 and 2 (LATS1/2), have been shown to decrease PD-L1 expression [66]. 

Amplification of *CD274* has been reported in up to 6% of nonsquamous NSCLC [16,70,71,72] and to be associated with high PD-L1 expression [16,71] as well as increased TMB [16,73]. These associations are remarkable as they might also predict susceptibility to anti-PD-1 treatment [74].

The integrity of 3′-UTR is critical for the regulation of *CD274* mRNA transcription and PD-L1 expression. In fact, disruption of 3′-UTR determines immune escape through increased PD-L1 expression in several cancer cell lines, including PC-9 lung adenocarcinoma cell lines [75]. 

### 2.2. Epigenetic Factors

DNA methylation and histone acetylation are epigenetic modifications that modulate gene expression in normal as well as a tumor cell. Aberrant epigenetic changes play a key role in cancer progression and development and can also affect PD-L1 expression [76]. 

#### 2.2.1. Methylation

Methylation of the *CD274* promoter downregulates PD-L1 transcription, and is reported to be implicated in resistance to anti-PD-L1 ICIs in NSCLC resistant to EGFR inhibition [77,78]. In lung cancer cells undergoing epithelial–mesenchymal transition (EMT), tumor growth factor-beta 1 (TGF-β1) decreases expression of the DNA methyl-transferase 1 (DNMT1), an enzyme that methylates the *CD274* promoter, thus resulting in increased PD-L1 expression [77]. Similarly, the addition of NSCLC cell lines of azacytidine, a general inhibitor of methylation, increases *CD274* transcription and PD-L1 expression [79]. Decitabine, an azacytidine-related hypomethylating agent, can also synergistically improve the activity of anti-PD-L1 ICIs in lung cancer cells, compared to either agent alone [80]. 

#### 2.2.2. Histone Modifications

In NSCLC, histone deacetylase (HDAC) 10 expression is positively correlated with PD-L1 expression and is independently associated with poor outcome [81]. Histone deacetylase inhibitors (HDACis) have been recently investigated as therapeutic agents in several cancer types and have been shown to require an intact immune system to exert their anticancer action [82]. HDACis increase PD-L1 expression and immune infiltrate in NSCLC models and have been shown to synergize with anti-PD-1 blockade in NSCLC models [83,84]. 

#### 2.2.3. Micro RNAs

MicroRNAs (miRNAs) are small non-coding RNAs that regulate target genes by inhibition of their translation after transcription, whose deregulation is associated with cancer genesis and progression [85]. Depending on the targeted mRNA, miRNAs can have different effects on PD-L1 expression. In NSCLC cell lines, miR-135 which targets tripartite-motif (TRIM) 16, and miR-3127-5p which positively regulates STAT3, are associated with increased PD-L1 expression [86,87]. On the other hand, miR-197 inhibits Cyclin-dependent kinases regulatory subunit 1 (CKS1B) by decreasing STAT3 activation thus reducing PD-L1 expression on NSCLC cells [88]. Similarly, miR-142-5p reduces PD-L1 expression through PTEN inhibition on lung adenocarcinoma cells [89]. *CD274* transcript can also be directly targeted through binding the 3′-UTR by miRNAs, such as miR-140, miR-200c, and miR-34, resulting in decreased *CD274* translation and, eventually, PD-L1 expression [50,90,91]. In addition, a signature of seven miRNAs found in the serum of NSCLC patients (215-5p, 411-3p, 493-5p, 494-3p, 495-3p, 548j-5p and 93-3p) has been associated with overall survival to nivolumab [92].

## 3. Extrinsic Factors 

Inducible expression of PD-L1 is mediated by extrinsic factors such as inflammatory signals, cytokines, growth factors, hypoxia, radiation therapy, chemotherapy and targeted therapies [43,93,94,95,96,97,98,99]. 

### 3.1. Cytokines

Chronic inflammation promotes the release of pro-inflammatory cytokines, such as Interferon-gamma (IFN-γ) and alfa (IFN-α), tumor necrosis factor-alfa (TNF-α), Interleukin (IL)-1a and IL-27, which induce PD-L1 expression in the TME [61,100,101,102]. IFN-γ released by tumor-infiltrating lymphocytes (TILs) is the most important cytokine in inducing PD-L1 expression on tumor cells, promoting tumor immune escape [103,104]. The IFN-γ-mediated PD-L1 upregulation occurs mainly through JAK/STAT/IRF1 signaling pathway activation in several types of cancers, including NSCLC, resulting in binding of IRF1 to the *CD274* promoter [61,100,105,106]. Similarly, TNF-α, IL-1a and IL-27 can induce PD-L1 expression, acting synergistically with IFN-γ [103,107]. On the contrary, IL-10 inhibits expression of IFN-γ targeted genes, including PD-L1 [108].

### 3.2. Growth Factors

The interaction between tumor cells and TME can modulate the effect that intrinsic factors exert on PD-L1 expression. For example, as already mentioned, EGFR activation by EGF binding can induce PD-L1 expression on tumor cells; however, *EGFR*-mutant NSCLCs tend to have lower PD-L1 expression levels compared to *EGFR*-wild type tumors as a result of the interaction between tumor cells and the immune TME [38,39,40].

TGF-β is a key inductor of EMT which is characterized by increased tumor proliferation, invasion, metastasis and immune surveillance escape ability [77,109,110]. It was demonstrated that TGF-β1-induced EMT leads to increased PD-L1 expression in NSCLC cell lines through both the demethylation of PD-L1 gene promoter and the activation of the NF-κB signaling pathway [109]. Based on this data, a combination of PD-L1 blockade and TGF-β inhibition represents an intriguing therapeutic strategy to overcome tumor immune escape and resistance, as discussed below. In cancer stem cells, and to a lesser extent in normal cancer cells, EMT-associated β-catenin activation induces expression of N-glycosyltransferase STT3 that stabilized and thus upregulates PD-L1 through N-glycosylation [111].

### 3.3. DNA Damaging Agents

Radiation therapy and chemotherapy can induce PD-L1 expression thanks to their ability to generate DNA damage [94]. Specifically, radiation therapy and chemotherapy induce DNA double-strand breaks as the main type of DNA damage whose repair is associated with PD-L1 upregulation in cancer cells through the activation of ATM/ATR/Checkpoint kinase 1 (Chk1) and STAT1/3–IRF1 pathway [94]. Recently in vitro study showed that treatment with pemetrexed enhanced PD-L1 expression, both in its membrane-bound form and its soluble form, in non-squamous NSCLC cell lines through activation of mTOR and STAT3 signaling pathways, whereas other chemotherapeutic agents (e.g., gemcitabine, paclitaxel, vinorelbine, cisplatin) did not influence PD-L1 levels [112]. In addition, pemetrexed led to increased release of pro-inflammatory cytokines (e.g., IFN-γ and IL-2), that further stimulated PD-L1 expression on tumor cells, resulting in an immune favorable TME for anti-PD-1/PD-L1 therapy. These results could explain the significant clinical benefit obtained with platinum-pemetrexed and pembrolizumab combination as upfront treatment also in the subgroup of metastatic NSCLC patients with PD-L1 negative tumors [5].

### 3.4. Targeted Therapies

Additionally, targeted therapies can affect PD-L1 expression levels in NSCLC. As above mentioned, *EGFR* mutations are commonly associated with PD-L1 upregulation in NSCLC. Inhibition of EGFR signaling by EGFR tyrosine kinase inhibitors (TKIs), such as gefitinib and osimertinib, decreased PD-L1 expression through both down-regulation of PD-L1 mRNA and increased PD-L1 proteasomal degradation [29,96,113]. This can potentially lead to enhanced antitumor immunity, although not all studies were concordant since treatment with TKIs led to PD-L1 upregulation, resulting in immune suppression according to some reports [114]. It was so hypothesized that TKIs could have a dual-phase regulation, with PD-L1 downregulation in an early phase, corresponding to the initial potent antitumor activity of these drugs, followed by a later phase, corresponding to the onset of TKI resistance, in which PD-L1 expression is increased [114,115]. Hence, monitoring changes in PD-L1 expression levels during EGFR TKI treatment could help predict the development of resistance [114].

### 3.5. Angiogenesis and Hypoxia

Cancer cells also produce proangiogenic factors that induce aberrant angiogenesis within tumors that is a crucial process for tumor growth and survival. The vascular endothelial growth factor (VEGF) is the most important proangiogenic factor, which also has pleiotropic effects on antitumor immune response since it inhibits antigen presentation, promotes regulatory T-cell infiltration and induces PD-L1 expression on tumor-infiltrated T-cells [116]. 

Tumoral aberrant angiogenesis caused abnormal vasculature which eventually generates hypoxia, a hallmark of TME. Cancer cells survive in hypoxic conditions through the activation of hypoxia-inducible factors (HIFs) [117]. Among HIFs, HIF-1α is a major oncogenic factor that facilitates the adaptation of cancer cells to stress, allowing their survival. HIF-1α directly binds the hypoxia-response element (HRE) in the *CD274* proximal promoter, inducing PD-L1 expression in myeloid-derived suppressor cells (MDSCs) and tumor cells [43,118]. Interestingly, lactates produced in hypoxic TME interact with their receptor GPR81 and induce PD-L1 expression through the activation of TAZ, which is a major downstream effector of the Hippo Pathway [69]. Consequently, hypoxia leads to increased HIF-1α levels resulting in T-cell suppression and an overall immunosuppressive TME [43,118,119]. 

## 4. Post-Translational Modification

PD-L1 expression and stability are finely tuned by several post-translational modifications, such as phosphorylation, glycosylation and ubiquitination [120,121]. 

### 4.1. Phosphorylation

The serine/threonine kinases glycogen synthase kinase 3beta (GSK3β) and AMP-activated protein kinase (AMPK) phosphorylate PD-L1 at the Thr180 and Ser184 residues, respectively, located in two specific and evolutionarily conserved phosphorylation motifs [122,123]. PD-L1 phosphorylation leads to increased PD-L1 E3 ligase mediated-proteasomal degradation, when PD-L1 is phosphorylated by GSK3β, while it leads to endoplasmic reticulum-associated degradation, when PD-L1 is phosphorylated by AMPK. Furthermore, EGF-mediated GSK3β inactivation prevents PD-L1 phosphorylation and poly-ubiquitination, stabilizing the protein and increasing its expression [117,122,124]. 

### 4.2. Glycosylation

Glycosylation is a crucial post-translational modification that alters protein conformation and stability, regulating their activities and interactions [124]. PD-L1 is highly glycosylated in human tumor tissues and in cancer cell lines [122]. Glycosylation of PD-L1 is completely inhibited by the N-linked glycosylation inhibitor tunicamycin, whereas it is not affected by O-glycosidase treatment, suggesting that N-glycosylation is the main type of PD-L1 glycosylation [122]. Furthermore, N-glycosylation occurs exclusively at evolutionarily conserved motifs (N35, N192, N200 and N219) in PD-L1 extracellular domain. N-glycosylation of N192, N200 and N219, but not N35, on PD-L1 protein, antagonizes the interaction between PD-L1 and GSK3β through steric hindrance and, consequently, prevents GSK3β-mediated phosphorylation of PD-L1 [117,124]. The resulting prevention of its ubiquitination and degradation by the 26S proteasome, stabilizes PD-L1 protein, supporting a key role of PD-L1 glycosylation in promoting tumor immune evasion [122,125]. Interestingly, the majority of PD-L1 in tumor cells is glycosylated, so that it cannot be degraded by the proteasome [121,122], and is masked to diagnostic immunohistochemistry assays [126]. 

### 4.3. Ubiquitination

Ubiquitination is an important post-translational modification, leading to proteasomal degradation of target proteins, including PD-L1 [127]. Ubiquitination is a key regulator of PD-L1 stability, through a balance between mono- and multi-ubiquitination on one side, and poly-ubiquitination on the other side [120]. Specifically, the attachment of ubiquitin to proteins as a monomer on one or more lysine residues is referred to as monoubiquitination or multiubiquitination, respectively, whereas its conjugation as a polymer, through sequential cycles of ubiquitination, leads to polyubiquitination [128]. Poly-ubiquitination of non-glycosylated PD-L1 leads to protein degradation and, consequently, to PD-L1 downregulation, whereas it was reported that PD-L1 mono- and multi-ubiquitination, e.g., in response to EGF stimulation, leads to increased PD-L1 protein expression [120]. 

### 4.4. Palmitoylation 

Palmitoylation (also known as thioacylation or S-acylation) is the main type of protein lipidation, consisting of the addition of palmitate to a cysteine residue, generally with a reversible thioester linkage [129]. Palmitoylation plays a crucial role in the regulation of trafficking and function of several proteins including RAS and EGFR, but also in the regulation of PD-L1 stability. In fact, palmitoylation of PD-L1, mainly catalyzed by palmitoyltransferase ZDHHC3 (*DHHC3*), leads to suppression of PD-L1 ubiquitination and degradation, resulting in protein stabilization, eventually promoting tumor immune escape [130]. Based on this data, targeting this post-translational modification, e.g., by silencing *DHHC3*, could represent a promising therapeutic strategy to promote PD-L1 degradation and activate antitumor immunity.

## 5. Potential Clinical Applications

The increasing understanding of the molecular mechanisms underlying PD-L1 expression regulation unravels potential therapeutic approaches to improve immunotherapy efficacy. 

### 5.1. Targeting the PI3K/AKT/mTOR Pathway

The PI3K/AKT/mTOR pathway is a key regulator of PD-L1 expression in NSCLC since it mediates both constitutive and inducible expression resulting from either alteration of oncogenes (e.g., *EGFR*, *KRAS*, and *ALK*) or extrinsic factors (e.g., IFN-γ or EGF), respectively.

Suppression of PI3Kγ–mTOR signal by PI3Kγ inhibition enhances tumor regression and prolongs survival in a synergistic fashion with PD-1 blockade in mouse models of cancer [47]. Buparlisib (BKM120) is an oral pan-PI3K inhibitor that was evaluated in phase II BASALT-1 trial in 63 pre-treated patients with recurrent NSCLC harboring documented aberrations in the PI3K pathway [131]. However, the study stopped early for futility because it did not meet its efficacy criterion (12-week PFS ≥ 50%) [131].

PI3K/Akt/mTOR pathway inhibition after progression on PD-1/PD-L1 blockade may overcome acquired resistance to ICIs secondary to PI3K/Akt/mTOR upregulation. To test this hypothesis, a phase study II trial of Ipatasertib, a novel highly selective ATP-competitive pan-Akt inhibitor, in combination with docetaxel in advanced NSCLC patients progressed to first-line immunotherapy is currently ongoing (NCT04467801). Trials of combination therapies of PI3K/Akt/mTOR inhibitors with immune checkpoint inhibitors are currently ongoing (Table 1).

### 5.2. Targeting DNA Damage Response

Among DDR proteins, poly (ADP-ribose) polymerase (PARP) is a family of protein which has been successfully targeted in breast, ovarian, prostate, and pancreatic cancers [132,133,134,135,136,137]. PARP inhibitors (PARPis), such as olaparib, talazoparib, rucaparib and niraparib, synergistically enhance PD-L1 blockade efficacy both in vitro and in vivo models [138,139,140,141] and are thus being evaluated in combination with ICIs in different cancer types [142,143,144]. No definitive data is currently available about combination treatments in NSCLC patients, but several trials are ongoing (Table 1).

### 5.3. Epigenetic Agents

Novel strategies involving drugs targeting epigenetic modifications as single-agent or in combination with other treatments have been developed in preclinical models and are moving to the clinical setting in NSCLC [145]. Combinations of epigenetic drugs and immunotherapies are currently being evaluated in multiple trials enrolling lung cancer patients. The ENCORE 601 study is a phase Ib/II trial of entinostat, an orally available class I selective HDACi, in combination with pembrolizumab in NSCLC, melanoma and mismatch-repair proficient colorectal cancer patients (NCT02437136) [146]. The NSCLC cohort included both anti-PD-1/PD-L1 naïve and pre-treated patients and enrolled 22 patients. In these patients, the most commonly reported grade 3/4 treatment-related adverse events were hypophosphatemia (9%), neutropenia (5%), anemia (5%), acute respiratory failure (5%), elevated alkaline phosphatase (5%) and immune-mediated hepatitis (5%). However, the combination showed a modest activity profile: the best response was stable disease in 3/6 pretreated patients, while one partial response, one stable disease and 9 progressive diseases were reported in PD-(L)1-naïve patients. 

Azacytidine is a DNA-methyltransferase inhibitor and was evaluated in phase II placebo-controlled trial (NCT02546986) in combination with pembrolizumab versus pembrolizumab alone as second-line treatment in advanced NSCLC (*N* = 100) [147]. Nonetheless, no significant difference in PFS between the experimental and the control arms (2.9 versus 4.0 months, respectively) was observed. In addition, the combination arm showed the worst tolerability profile, mainly for gastrointestinal adverse events.

HDACi and bromodomain and extra-terminal (BET) inhibitors are emerging in preclinical models as target agents with the potential to increase the effectiveness of ICIs and were recently evaluated in cancer trials [83,148]. Birabresib (OTX015) and INCB057643 are both BET inhibitors and their safety and efficacy as monotherapy or combination with standard of care in solid tumors are currently being investigated in clinical trials, also enrolling NSCLC patients, such as NCT02259114 and NCT02711137 for OTX015 and INCB057643, respectively. The most promising of these studies is a phase I study of birabresib in advanced solid tumors that enrolled 46 patients, including 10 NSCLC patients [149]. The trial aimed to identify the best phase II dose schedule among the two tested, i.e., 80 mg on a daily schedule and 100 mg for 7 consecutive days every 21-day cycle. The safety profile of the selected regimen (80 mg daily administered to *N* = 290 patients) was manageable, and included thrombocytopenia (*N* = 5), anemia (*N* = 2), and transaminase elevation (*N* = 1) as most commonly reported grade 3 or higher toxicities.

### 5.4. Targeting Growth Factors

Among extrinsic factors affecting PD-L1 expression, TGF-β1 has been investigated as an appealing potential target to improve immunotherapy efficacy. M7824 is a first-in-class bifunctional fusion protein, consisting of the anti-PD-L1 antibody avelumab fused with two extracellular domains made of a TGF-β receptor (TGF-βRII), which functions as a TGF-β “trap” [150]. M7824 has the ability to both inhibit PD-1/PD-L1 interaction and block TGF-β mediated mesenchymalization and tumor immune suppression, resulting in enhanced antitumor activity [150,151]. Safety and activity of M7824 were evaluated in a phase I trial enrolling 80 patients with advanced NSCLC patients progressing after first-line treatment [152]. Updated results of the 1200 mg dose cohort showed an objective response rate of 27.5%, with a median duration of response of 18 months and a median survival of 21.7 months in PD-L1 positive (≥1%) patients, together with a manageable safety profile. A phase III trial investigating the safety and efficacy of M7824 compared to pembrolizumab as first-line treatment of advanced NSCLC patients with high PD-L1-tumor expression is currently ongoing (NCT03631706). VEGF pathway blockade results in normalization of tumor vasculature and reduced hypoxia, increasing T-cell tumor infiltration, which can establish an immune-permissive TME. According to these data, anti-VEGF treatment can act synergistically with ICIs, enhancing immune response. Preclinical data on Lewis lung carcinoma mouse models demonstrated a synergic effect by the combination of anti-PD-1 and endostar (a recombinant humanized endostatin, anti-angiogenic molecule) in suppressing tumor growth by means of improving the TME and activating PI3K/AKT/mTOR-mediated autophagy [153]. The phase III IMpower150 study randomized chemotherapy-naïve patients with advanced NSCLC to receive atezolizumab plus carboplatin plus paclitaxel (ACP group), atezolizumab plus bevacizumab plus carboplatin plus paclitaxel (ABPC group) or bevacizumab plus carboplatin plus paclitaxel (BCP) [154]. This study showed that the ABPC regimen compared to the BCP one provided a significant improvement in terms of ORR (56% vs. 40%, respectively), median PFS (8.4 vs. 6.8 months, respectively), and median OS (19.8 vs. 14.9 months, respectively).

### 5.5. Targeting Post-Translational Modifications

Post-translational modifications that modulate PD-L1 expression, such as glycosylation, phosphorylation, and ubiquitination, could be potential targets for anti-cancer treatment. COP9 signalosome 5 (CSN5) is involved in TNF-a-mediated PD-L1 stabilization because of its function in deubiquitination of PD-L1 and its inhibition by curcumin may benefit immunotherapy in the reduction of tumor growth such as demonstrated in in vivo preclinical mouse models treated with CTLA-4 antibodies [155].

## 6. Conclusions

PD-L1 expression in NSCLC is highly variable because of the interaction of many factors, that include intrinsic (genetic, epigenetic, and post-translational factors) and extrinsic factors. The understanding of these factors may unravel potential targets to improve outcomes on ICIs in these patients, which are under investigation in clinical trials.

## Figures and Tables

**Figure 1 cancers-12-03129-f001:**
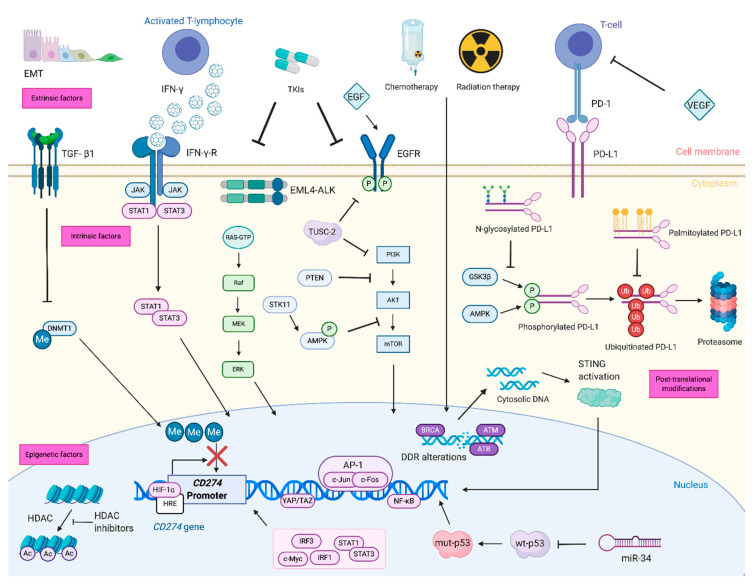
Regulation of PD-L1 expression involved several factors at different levels. The main extrinsic factors involved in PD-L1 regulation are pictured on the top of the picture and include cytokines (e.g., IFN-γ), growth factors (e.g., EGF), hypoxia, chemotherapy, radiation therapy and TKIs, which regulate PD-L1 expression on the transcriptional level. Among intrinsic factors, genetic alterations, involve intracellular signaling pathways (e.g., MAPK, PI3K/Akt/mTOR, JAK/STAT), tumor suppressor genes (e.g., *TP53*, *STK11*), DDR genes (e.g., *BRCA1-2*, *ATM*, *ATR*) and transcription factors (e.g., c-Myc, NF-κB). Other intrinsic factors include epigenetic modifications: *CD274* promoter methylation, inhibition of histone deacetylase and regulation of gene translation by miRNAs. The main post-translational modifications able to regulate PD-L1 stability and expression are phosphorylation, ubiquitination, glycosylation and palmitoylation (pictured on the right-hand side). Created with BioRender.com. EGF: Epidermal Growth Factor; EGFR: Epidermal Growth Factor receptor; EML4-ALK: echinoderm microtubule-associated protein-like 4 gene-anaplastic lymphoma kinase gene; IFN-γ: interferon-gamma; IFN-γR: interferon-gamma receptor; TGF-β1: tumor growth factor beta 1; TKIs: tyrosine kinase inhibitors; VEGF: vascular endothelial growth factor; PD-1: programmed-cell death; PD-L1: programmed-cell death ligand 1; TUSC2: tumor suppressor candidate 2; PI3K: phosphatidylinositol 3-kinase; Akt: Protein kinase B; mTOR: mammalian target of rapamycin; JAK: Janus kinase; STAT: signal transducer and activator of transcription; NF-κB: nuclear factor kappa-light-chain-enhancer of activated B cells; TP53: Tumor protein p53; *STK11*: Serine/threonine kinase 11; DDR: DNA damage response and repair; *BRCA1-2*: breast cancer type 1 and 2 susceptibility proteins; *ATM*: Ataxia Telangiectasia Mutated; *ATR*: *ATM* related protein; RAS: Rat Sarcoma Viral Oncogene Homolog; MEK: Mitogen-activated ERK kinase; ERK: Extracellular signal-Regulated Kinases; AMPK: 5′ AMP-activated protein kinase; PTEN: Phosphatase and tensin homolog; AP1: Activator protein 1; DNMT1: DNA methyl-transferase 1; IRF: interferon regulatory factor; Me: methyl group, P: phosphate group, *STING*: stimulator of interferon genes; HIF-1α: Hypoxia-inducible factor 1-alpha, HRE: hypoxia-response element, HDAC: Histone deacetylase; HDACi: Histone deacetylase inhibitors; EMT: epithelial–mesenchymal transition; GSK3β: glycogen synthase kinase 3beta.

**Table 1 cancers-12-03129-t001:** Ongoing clinical trials of therapies targeting PD-L1 expression regulation, in combination with immune checkpoint inhibitors (ICIs) in non-small-cell lung cancer (NSCLC): (source: www.clinicaltrials.gov, last accessed: 9 September 2020).

NCT	Phase	Experimental Agent	Tumor	Setting	*N*	Arm(s)	Primary Outcome(s)
**PI3K Inhibitor**							
NCT02637531	I	IPI-549	AST	After standard	219	IPI-549 ± Nivolumab	AEs DLT
NCT03257722	Ib/II	Idelalisib	NSCLC	Progressed on chemotherapy and ICIs	40	Idelalisib + Pembrolizumab	DLT
NCT04282018	I/II	BGB-10188	AST	After standard	150	BGB-10188 + Tislelizumab	AEs
**mTOR Inhibitor**							
NCT04348292	I	Sirolimus	Stage I-IIIA NSCLC	Neoadjuvant	31	Sirolimus + Durvalumab	AEs CPRR
NCT03190174	I/II	ABI-009 (Nab-rapamycin)	AST	Any line	40	ABI-009 + Nivolumab	MTD
**JAK1 Inhibitor**							
NCT03425006	II	Itactinib	NSCLC	1st line	48	Itactinib + Pembrolizumab	ORR AEs
**PARP Inhibitor**							
NCT03330405	Ib/II	Talazoparib	AST	*BRCA* or *ATM* deficient	214	Talazoparib + Avelumab	DLT ORR
NCT03559049	I/II	Rucaparib	NSCLC	Maintenance after 1st line immuno-chemotherapy	55	Pembrolizumab + Rucaparib	PFS
NCT04538378	II	Olaparib	EGFR-mutated NSCLC transformed to SCLC	After platinum-based chemotherapy ± immunotherapy for SCLC trasformation	14	Olaparib + Durvalumab	BOR
NCT03775486	II	Olaparib	NSCLC	Maintenance after 1st line chemotherapy	401	Olaparib ± Durvalumab	PFS
NCT03308942	II	Niraparib	NSCLC	1^st^ line	53	Niraparib ± Pembrolizumab/Dostarlimab	ORR
NCT04380636	III	Olaparib	Stage III NSCLC	Unresectable	870	Pembrolizumab + chemoradiation→ Pembrolizumab ± OlaparibChemoradiation→Durvalumab	PFS OS
NCT03976323	III	Olaparib	NSCLC	Maintenance after 1st line immuno-chemotherapy	792	Pembrolizumab + Olaparib Pembrolizumab + Pemetrexed	PFS OS
NCT03976362	III	Olaparib	NSCLC	Maintenance after 1st line immuno-chemotherapy (CBDCA + taxane)	735	Pembrolizumab ± Olaparib	PFS OS
***ATR* Inhibitor**							
NCT03334617	II	AZD6738	NSCLC	≥2nd line	340	Durvalumab + AZD6738	ORR
NCT03833440	II	AZD6738	NSCLC	3rd - 4th line (after ICI)	120	Durvalumab + AZD6738	DCR at 12 weeks
**TGFBR1 Inhibitor**							
NCT03732274	I/II	Vactosertib (TEW-7197)	NSCLC	After chemotherapy	63	Vactosertib + Durvalumab	MTD
**Epigenetic Drugs**							
NCT02437136	I/II	Entinostat	NSCLC, melanoma, mismatch repair-proficient CRC	≥2nd line	202	Entinostat + Pembrolizumab	AEs ORR
NCT01928576	II	Azacytidine	NSCLC	Any line	120	Azacitidine + Entinostat + Nivolumab	ORR
NCT02546986	II	Azacytidine	NSCLC	2nd line	100	Pembrolizumab ± Azacitidine	PFS
NCT04250246	II	Guadecitabine	NSCLC Melanoma	≥2nd line (only one prior line of ICIs allowed)	184	Ipilimumab + Nivolumab ± Guadecitabine	ORR

N: planned sample size; NCT: clinicaltrials.gov identifier; AEs: adverse events; AST: advanced solid tumors; BOR: best overall response; CPRR: complete pathological response rate; CRC: colorectal cancer; DCR: disease control rate; DLT: dose-limiting toxicities; MTD: maximum tolerated dose; NSCLC: non-small-cell lung cancer; ORR: objective response rate; OS: overall survival; PFS: progression-free survival.

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
