# Peer review of "The Mechanisms of PD-L1 Regulation in Non-Small-Cell Lung Cancer (NSCLC): Which Are the Involved Players?"

_cancers, 2020, doi:10.3390/cancers12113129_

Round 1

Reviewer 1 Report

This is a very interesting and well written review describing mechanisms and players regulating PD-L1 expression with a focus on NSCLC. The work is well structured and comprehensively discusses the different aspects of PD-L1 expression on tumor cells: both intrinsic (genetic alterations and epigenetic factors) and extrinsic factors (i.e. cytokines, external agents or specific conditions) are initially described. The authors than reported the modifications that occurred at the posttranscriptional level such as glycosylation, phosphorilation, ubiquitination and palmitoylation. They finally conclude with the clinical implication and future prospective to improve patients outcome.

Author Response

This is a very interesting and well written review describing mechanisms and players regulating PD-L1 expression with a focus on NSCLC. The work is well structured and comprehensively discusses the different aspects of PD-L1 expression on tumor cells: both intrinsic (genetic alterations and epigenetic factors) and extrinsic factors (i.e. cytokines, external agents or specific conditions) are initially described. The authors than reported the modifications that occurred at the posttranscriptional level such as glycosylation, phosphorilation, ubiquitination and palmitoylation. They finally conclude with the clinical implication and future prospective to improve patients outcome.

 We thank the reviewer for these comments.

Reviewer 2 Report

The current study by Lamberti et al., entitled "The mechanisms of PD-L1 regulation in non-small cell lung cancer (NSCLC): which are the involved players?" is an interesting review paper focusing on mechanisms implicated in the regulation of PD-L1 expression in NSCLC as well as on their potent therapeutic exploitation. Despite the fact that the authors have interpreted the available studies with a very informative way, providing a useful figure and a comprehensive table, there are some issues with this manuscript, which need to be treated:

  1. Please, provide expansions and explanations of the used abbreviations and acronyms.
  2. The manuscript needs to be extensively edited by a native English speaker in order to improve some obscure points.
  3. Commas should be used appropriately throughout the manuscript
  4. p2, line 49. The word “viable” should be set before tumor cells regarding the definition of TPS.
  5. p2, line 52: Provided references refer to pembrolizumab, which targets PD-1 and not PD-L1.
  6. p2, line 65-69. Please, restructure this sentence clarifying the meaning based on the provided references. Now, the meaning of the sentence is obscure
  7. p2, lines 65-89. This huge paragraph could be improved if it was separated to more paragraphs. It addresses many different issues (KRAS, EGFR etc)
  8. p2, lines 82-86, The sentence needs restructure.
  9. p2, line 73. Please, restructure the first part of this sentence.
  10. p2, line 89. Please provide a reference.
  11. Lines 100-146. This paragraph needs restructure. The current form makes confusion. Better organization of the presented data as well as more paragraphs are needed.
  12. p4, line 147. The genes should be presented in italics. The authors should decide which of the symbols for PD-L1/CD247 gene they will use throughout of the manuscript.
  13. p4, line158-163. Please, rephrase.
  14. p4, line 166. This paragraph is particularly deficient since it doesn’t provide any data regarding the role of histone modifications in the PD-L1 regulation. In addition, more data regarding the role of HDACs in PDL1 expression could be presented.
  15. In Figure 1, please incorporate the provided expansions for the used abbreviations and acronyms in the figure legend.
  16. p5, line 191. In NF-kB, k should be replaced with the Greek “κ” according to the initial publication of Baltimore and Sen.
  17. p6, line 214. Please, replace “target therapy” with “targeted therapy”
  18. p6, lines 225-226. Please, rephrase.
  19. p6, lines 229-230. This sentence is irrelevant with the impact of “growth factors” in PDL1 regulation
  20. p6, line 244. Which chemotherapeutic agents?
  21. p6, line 251. Replace “target” with “targeted”.
  22. p7, lines 269-270. Please rephrase.
  23. p7, line 275. Please, explain the “TAZ-dependent way”
  24. p7, lines 276-277. This sentence doesn’t provide any information regarding the PD-L1.
  25. p7, line 279-281, please, rephrase.
  26. p8, lines 312-314. This paragraph has only one sentence. It should be incorporated in the previous.
  27. p8, line 319. The authors should use the same symbol for the same protein. (e.g Ras).
  28. p8, lines 336-337. Please, rephrase; Provide reference.
  29. p9, lines 352-354. This statement is not true since drugs which target epigenetic modifications are already in clinical practice.

Author Response

The current study by Lamberti et al., entitled "The mechanisms of PD-L1 regulation in non-small cell lung cancer (NSCLC): which are the involved players?" is an interesting review paper focusing on mechanisms implicated in the regulation of PD-L1 expression in NSCLC as well as on their potent therapeutic exploitation. Despite the fact that the authors have interpreted the available studies with a very informative way, providing a useful figure and a comprehensive table, there are some issues with this manuscript, which need to be treated:

Point 1: Please, provide expansions and explanations of the used abbreviations and acronyms.

Response 1: We thank the reviewer for these suggestions. Explanations for the used abbreviations and acronyms (e.g. MAPK, PTEN) have been added in the text.

Point 2: The manuscript needs to be extensively edited by a native English speaker in order to improve some obscure points.

Point 3: Commas should be used appropriately throughout the manuscript.

Response 2+3: We thank the reviewer for this comment. The manuscript is thoroughly revised.

Point 4: p2, line 49. The word “viable” should be set before tumor cells regarding the definition of TPS.

Response 4: We thank the reviewer for this suggestion. We have added the word “viable” in the definition of TPS (line 80).

Point 5: p2, line 52: Provided references refer to pembrolizumab, which targets PD-1 and not PD-L1.

Response 5: We thank the reviewer for this comment. Sentence it was corrected accordingly (line 84).

Point 6: p2, line 65-69. Please, restructure this sentence clarifying the meaning based on the provided references. Now, the meaning of the sentence is obscure.

Response 6: We thank the reviewer for this comment. Sentence has been rephrased accordingly (line 99-105).

Point 7: p2, lines 65-89. This huge paragraph could be improved if it was separated to more paragraphs. It addresses many different issues (KRAS, EGFR etc)

Response 7: We thank the reviewer for this comment. We divided the paragraph further.

Point 8: p2, lines 82-86, The sentence needs restructure.

Response 8: We thank the reviewer for this comment. The sentence was rephrased accordingly (lines 126-132).

Point 9: p2, line 73. Please, restructure the first part of this sentence.

Response 9: We thank the reviewer for this comment. The sentence was rephrased accordingly (lines 113-117).

Point 10: p2, line 89. Please provide a reference.

Response 10: We added the needed reference (Koh J et al 2016) (lines 214).

Point 11: Lines 100-146. This paragraph needs restructure. The current form makes confusion. Better organization of the presented data as well as more paragraphs are needed.

Response 11: We thank the reviewer for this suggestion, we edited the paragraph accordingly (lines 208-268).

Point 12: p4, line 147. The genes should be presented in italics. The authors should decide which of the symbols for PD-L1/CD247 gene they will use throughout of the manuscript.

Response 12: We thank the reviewer for these suggestions. We have used CD247 as symbol and we have used it throughout the manuscript.

Point 13: p4, line158-163. Please, rephrase.

Response 13: We thank the reviewer for this suggestion, we rephrased the sentence in a more direct way (lines 249-254).

Point. 14: p4, line 166. This paragraph is particularly deficient since it doesn’t provide any data regarding the role of histone modifications in the PD-L1 regulation. In addition, more data regarding the role of HDACs in PDL1 expression could be presented.

Response 14: We thank the reviewer for this suggestion, we improved this paragraph with additional data. However, unluckily, we recognize that robust data about HDAC and PD-L1 expression are lacking (lines 258-264).

Point 15: In Figure 1, please incorporate the provided expansions for the used abbreviations and acronyms in the figure legend.

Response 15: We thank the reviewer for these comments. Figure legend has been revised accordingly, introducing the acronyms for EGF and MEK that were missing.

Point 16: p5, line 191. In NF-kB, k should be replaced with the Greek “κ” according to the initial publication of Baltimore and Sen.

Response 16: We thank the reviewer for this suggestion. We replaced the letter with the Greek symbol “κ” accordingly.

Point 17: p6, line 214. Please, replace “target therapy” with “targeted therapy”

Response 17: We thank the reviewer for this comment. We made the suggested correction (line 322).

Point 18: p6, lines 225-226. Please, rephrase.

Response 18: We thank the reviewer for this suggestion, we rephrased the sentence in a more direct way (lines 336-337).

Point 19: p6, lines 229-230. This sentence is irrelevant with the impact of “growth factors” in PDL1 regulation

Response 19: We thank the reviewer for this comment, we sentence has been removed (line 341).

Point 20: p6, line 244. Which chemotherapeutic agents?

Response 20: We thank the reviewer for this comment. We have specified the chemotherapeutic agents used in the study (line 398-399).

Point 21: p6, line 251. Replace “target” with “targeted”.

Response 21: We thank the reviewer for this comment. The error has been revised as requested.

Point 22: p7, lines 269-270. Please rephrase.

Response 22: We thank the reviewer for this suggestion, we rephrased the sentence in a more direct way (lines 428-429).

Point 23: p7, line 275. Please, explain the “TAZ-dependent way”

Response 23: We thank the reviewer for this comment. We rephrased the sentence to clarify this concept (lines 436-441).

Point 24: p7, lines 276-277. This sentence doesn’t provide any information regarding the PD-L1.

Response 24: We thank the reviewer for this comment. The sentence provides a limited outlook on the immune TME which is critical in determining inducible PD-L1 expression.  

Point 25: p7, line 279-281, please, rephrase.

Response 25: We thank the reviewer for this suggestion, we rephrased the sentence accordingly (lines 444-445).

Point 26: 8, lines 312-314. This paragraph has only one sentence. It should be incorporated in the previous.

Response 26: We thank the reviewer for this suggestion, we modified the sentence accordingly (lines 485-489).

Point 27: p8, line 319. The authors should use the same symbol for the same protein. (e.g Ras).

Response 27: We thank the reviewer for this suggestion, we reviewed accordingly (line 494).

Point 28: p8, lines 336-337. Please, rephrase; Provide reference.

Response 28: We thank the reviewer for this suggestion, we rephrased the sentence accordingly and provided the respective reference Vansteenkiste, JF - J. Thorac. Oncol. 2015 (lines 516-517).

Point 29: p9, lines 352-354. This statement is not true since drugs which target epigenetic modifications are already in clinical practice.

Response 29: We thank the reviewer for this comment. According to available literature, targeted therapies against epigenetic modifications are not already a standard treatment in clinical practice among NSCLC patients. We specified that it referred to NSCLC clinical practice (line 540-542).

Reviewer 3 Report

This is a well written and comprehensive review of intrinsic and extrinsic factors that are associated with PD-L1 levels in non small cell lung cancer. PD-L1, the ligand for PD-1 is an important modulator of T -cell mediated adaptive immune responses, which can be an important anti tumour mechanism and as such involved in mechanisms of immune evasion by tumours, typically through over expression. This in turn has led to successful targeted therapies exploiting either PD-1 or PD-L1. One aim of the review is to set the scene for better understanding how expression levels are regulated so that the mechanisms involved can be the subject of new therapies that either enhance immune specific anti tumour activities are reduce immune blockade by the tumour. In this regard it has largely succeeded. Every aspect from cell signalling, in particular the RAS/ MAPK pathways, PI3K/Akt/mTOR, tumour suppressors, epigenetic modification, effects on transcription and post translational targeting and modification are considered. Extrinsic factors include cytokines, growth factors and the microenvironment, in particular the roles of angiogenesis and hypoxia. the authors include extrinsic factors brought in as a consequence of therapy, for example radiation, DNA damaging agents and targeted therapies, especially of the latter, those that impact on the above.

There are several points of relevance to consider. One is the interdependence of many of the factors described, which raises which, if any is of prevailing importance when more than one is acting in given situation. Also, many of the factors are associated with particular PD-L1 levels, the actual mechanisms involved and again whether there is interdependence is typically less clear (hence the review) but it would be useful to give some insight as to specific mechanisms, (some are mentioned), whether known, relative importance or if not known, likely relative prioritisation and why. Given the therapeutic aims, it would also be interesting to predict the consequences of manipulating specific factors.

The subject is a hot topic and similar reports are in the latest literature but this review would have a place alongside. References selected are current, relevant and high impact. The collection is of value in itself.

The use of English is good but a little unnatural in a few places. This leads to a little confusion in places or inappropriate use of singulars/plurals, for example:

"Given its crucial role, PD-L1 expression on cell membrane are finely regulated at different levels."

"As opposite to this,"

neutrophil cf neutrophils.

TMB does seem to be defined and not all readers would necessarily know what is meant by it.

Author Response

Response to Reviewer 3 Comments

This is a well written and comprehensive review of intrinsic and extrinsic factors that are associated with PD-L1 levels in non small cell lung cancer. PD-L1, the ligand for PD-1 is an important modulator of T -cell mediated adaptive immune responses, which can be an important anti tumour mechanism and as such involved in mechanisms of immune evasion by tumours, typically through over expression. This in turn has led to successful targeted therapies exploiting either PD-1 or PD-L1. One aim of the review is to set the scene for better understanding how expression levels are regulated so that the mechanisms involved can be the subject of new therapies that either enhance immune specific anti tumour activities are reduce immune blockade by the tumour. In this regard it has largely succeeded. Every aspect from cell signalling, in particular the RAS/ MAPK pathways, PI3K/Akt/mTOR, tumour suppressors, epigenetic modification, effects on transcription and post translational targeting and modification are considered. Extrinsic factors include cytokines, growth factors and the microenvironment, in particular the roles of angiogenesis and hypoxia. the authors include extrinsic factors brought in as a consequence of therapy, for example radiation, DNA damaging agents and targeted therapies, especially of the latter, those that impact on the above.

There are several points of relevance to consider. One is the interdependence of many of the factors described, which raises which, if any is of prevailing importance when more than one is acting in given situation. Also, many of the factors are associated with particular PD-L1 levels, the actual mechanisms involved and again whether there is interdependence is typically less clear (hence the review) but it would be useful to give some insight as to specific mechanisms, (some are mentioned), whether known, relative importance or if not known, likely relative prioritisation and why. Given the therapeutic aims, it would also be interesting to predict the consequences of manipulating specific factors.

The subject is a hot topic and similar reports are in the latest literature but this review would have a place alongside. References selected are current, relevant and high impact. The collection is of value in itself.

Point 1: The use of English is good but a little unnatural in a few places. This leads to a little confusion in places or inappropriate use of singulars/plurals, for example:

"Given its crucial role, PD-L1 expression on cell membrane are finely regulated at different levels."

"As opposite to this,"

neutrophil cf neutrophils.

Response 1: We thank the reviewer for these comments. We made the suggested corrections (lines 69-70, line 109 and line 209).

Point 2: TMB does seem to be defined and not all readers would necessarily know what is meant by it.

Response 2: We thank the reviewer for this suggestion. We have added definition of TMB (lines 129-131).

Round 2

Reviewer 2 Report

The authors have addressed the majority of the issues. However, some minor points remain to be clarified.

In the second version of their submission three new authors have been added. Is this change approved by all authors? Have you informed the editor on this?

Gene names should be italicized throughout the manuscript (e.g. line 84, KRAS; line 119, ALK etc)

lines 175-176. The word oncogene should be deleted. The authors refer to the Myc protein and not to the gene.  

lines 204-205: This sentence should be modified, since this section refers to the epigenetic modifications in the context of PD-1/PDL-1 regulation.

line 207: Please, replace “downregulate” with “downregulates

Author Response

The authors have addressed the majority of the issues. However, some minor points remain to be clarified.

Point 1: In the second version of their submission three new authors have been added. Is this change approved by all authors? Have you informed the editor on this?

Response 1: We thank the reviewer for this comment. The addition of the three new authors had been acknowledged to other authors and to the Editor as per Journal procedures.

Point 2: Gene names should be italicized throughout the manuscript (e.g. line 84, KRAS; line 119, ALK etc)

Response 2: We thank the reviewer for this comment. We made the suggested corrections in the text.

Point 3: lines 175-176. The word oncogene should be deleted. The authors refer to the Myc protein and not to the gene.  

Response 3: We thank the reviewer for this suggestion, we reviewed accordingly.

Point 4: lines 204-205: This sentence should be modified, since this section refers to the epigenetic modifications in the context of PD-1/PDL-1 regulation.

Response 4: We thank the reviewer for this comment. Sentence it was corrected accordingly (lines 203-206).

Point 5: line 207: Please, replace “downregulate” with “downregulates

Response 5: We thank the reviewer for this comment. We made the suggested correction.